# Insight on Non-Coding RNAs from Biofluids in Ovarian Tumors

**DOI:** 10.3390/cancers15051539

**Published:** 2023-02-28

**Authors:** Yohann Dabi, Amélia Favier, Léo Razakamanantsoa, Léa Delbos, Mathieu Poilblanc, Philippe Descamps, Francois Golfier, Cyril Touboul, Sofiane Bendifallah, Emile Daraï

**Affiliations:** 1Department of Obstetrics and Reproductive Medicine, Hôpital Tenon, Sorbonne University, 4 Rue de la Chine, 75020 Paris, France; 2Department of Radiology Imaging and Interventional Radiology, Hôpital Tenon, Sorbonne University, 4 Rue de la Chine, 75020 Paris, France; 3Department of Obstetrics and Reproductive Medicine—CHU d’Angers, 49100 Angers, France; 4Department of Obstetrics and Reproductive Medicine, Lyon South University Hospital, Lyon Civil Hospices, 69495 Pierre-Bénite, France

**Keywords:** ovarian tumor, non-coding RNA, borderline ovarian tumor, ovarian cancer

## Abstract

**Simple Summary:**

Ovarian cancer is the most lethal gynecologic cancer since it is often diagnosed at advanced stages. Current tools for diagnosis are currently insufficient and include physical examination, ultrasound and pelvic magnetic resonance imaging, as well as algorithms combining thoraco-abdomino-pelvic scans and blood markers. In this context, there is a need for new tools not only to assess the diagnosis but also to predict the response to chemotherapy and to detect recurrences. Previous studies have highlighted the potential value of non-coding RNAs (ncRNA) in tissue samples, but rarely in biofluids. In this review, we aim to summarize the existing literature on ncRNAs and ovarian tumors in biofluids. Most studies are focused on serum and blood with no data on other biofluids and with few ncRNAs investigated using qRT-PCR or microarray, which does not reflect the heterogeneity of ovarian cancers.

**Abstract:**

Ovarian tumors are the most frequent adnexal mass, raising diagnostic and therapeutic issues linked to a large spectrum of tumors, with a continuum from benign to malignant. Thus far, none of the available diagnostic tools have proven efficient in deciding strategy, and no consensus exists on the best strategy between “single test”, “dual testing”, “sequential testing”, “multiple testing options” and “no testing”. In addition, there is a need for prognostic tools such as biological markers of recurrence and theragnostic tools to detect women not responding to chemotherapy in order to adapt therapies. Non-coding RNAs are classified as small or long based on their nucleotide count. Non-coding RNAs have multiple biological functions such as a role in tumorigenesis, gene regulation and genome protection. These ncRNAs emerge as new potential tools to differentiate benign from malignant tumors and to evaluate prognostic and theragnostic factors. In the specific setting of ovarian tumors, the goal of the present work is to offer an insight into the contribution of biofluid non-coding RNAs (ncRNA) expression.

## 1. Introduction

Adnexal masses represent a wide spectrum of tumors of various origins (ovary, fallopian tube, and pelvic organs), among them ovarian tumors are the most frequent, raising diagnostic and therapeutic issues linked to a large spectrum of tumors with a continuum from benign to malignant. The true incidence of ovarian tumors in the general population is unknown as most of them are asymptomatic and hence undiagnosed [1]. It is estimated that 10% of women will undergo surgery for an ovarian mass in their lifetime [2].

Ovarian tumors are generally detected at physical examination or at pelvic imaging for various reasons in asymptomatic patients. Less frequently, an ovarian tumor can be the source of symptoms, acute pain (torsion of the adnexa), or chronic pelvic pain related to compression of neighboring organs [1]. Occasionally, the diagnosis can be made in the context of ovarian cancer (OC) often diagnosed at an advanced stage alongside ascites, bloating, weight loss and peritoneal carcinomatosis [3,4].

OC is the fifth most common cancer and the most lethal gynecologic malignancy, with 313,959 new cases per year and 207,252 deaths per year in 2020 [5]. Epithelial ovarian cancer (EOC) represents more than 95% of all OC [6]. During their lifetime, approximately one in seventy women will develop the disease. The median age at diagnosis is 68 years with a maximum incidence in women in their 70s. The disease behaves as a chronic condition with relapses, and iterative chemotherapies can therefore lengthen survival [7]. The disease survival remains poor at 40% at five years, and 32% at ten years, since more than 75% of patients are diagnosed at advanced stage disease, while the five-year survival rate for women diagnosed at an early stage reaches 90%, underlining the potential benefit of biomarkers of early stages as well as for detecting the transition of a borderline tumor into invasive cancer [5].

Except for patients with deleterious mutation, for whom risk-reducing surgery is recommended, no screening for OC in the general population has proved its relevance [8]. In routine practice, first-line transvaginal ultrasound is used to differentiate benign, borderline and malignant ovarian tumors [9]. Van Calster et al. evaluated the clinical utility of six prediction models for ovarian malignancy [10], and found that the ADNEX models with and without cancer antigen 125 (CA125) determination and SRRisk were the best calibrated. However, previous studies have underlined that 18% to 31% of ovarian tumors remain indeterminate after ultrasonography using International Ovarian Tumor Analysis (IOTA) Simple Rules or other ultrasonography scoring systems [9,11,12]. MRI is the second-line imaging technique for characterizing ovarian tumors. Ovarian-Adnexal Reporting Data System Magnetic Resonance Imaging (O-RADS MRI) score consists of five categories according to the positive likelihood ratio for a malignant neoplasm [13] with a sensitivity of 0.93 and a specificity of 0.91. However, these results are based on an observational study without randomization, and the score was not integrated into clinical decision-making, limiting its utility.

CA125 is the most used biomarker for determining the nature of ovarian tumors [14,15], although normal levels have been reported in as many as 50% of early stage ovarian cancers [16,17,18]. A recent Cochrane review evaluated several algorithms to assess the risk of malignancy of ovarian tumors, including biological markers and imaging [19]. However, none of these scoring systems had sufficient relevance to characterize ovarian tumors [19].

Finally, a recent review by Funston et al. analyzing 18 documents from 11 countries showed that transabdominal/transvaginal ultrasound and the CA125 were the most widely advocated initial tests [20]. However, no consensus exists on the best strategy to improve diagnostic performance: “single test”, “dual testing”, “sequential testing”, “multiple testing options” and “no testing”. This further underlines the need for new biological tools to diagnose ovarian cancer in the general population as early as possible. In addition, there is a need for prognostic tools such as biological markers of recurrence and theragnostic tools to detect women not responding to chemotherapy in order to adapt therapies.

Among ncRNAs, those with less than 50 nucleotides are defined as small RNAs (sncRNAs) and those with more than 200 nucleotides are defined as long non-coding RNAs (lncRNAs). SncRNAs are further classified into microRNAs (miRNAs), Piwi interacting RNAs (piRNAs), transfer RNAs (tRNAs), small nuclear RNAs (snRNAs), and small interfering RNAs (siRNAs) [21,22]. LncRNAs are classified into intergenic ncRNAs (lincRNAs), some circular RNAs (circRNAs), and ribosomal RNAs (rRNAs) [23,24]. Numerous studies indicate that ncRNAs, representing 98% of the transcriptome, are essential for tumorigenesis by regulating the expression of tumor-related genes [25,26,27,28,29,30,31,32]. These ncRNAs emerge as new potential tools to differentiate benign from malignant tumors and to evaluate prognostic and theragnostic factors. In the specific setting of ovarian tumors, the goal of the present work is to offer an insight into the contribution of biofluid non-coding RNAs (ncRNA) expression.

## 2. miRNAs

miRNAs are small intracellular RNAs, 22 nucleotides long, capable of inducing the silencing of gene expression by post-transcriptional regulatory mechanisms [33] or alternatively by binding miRNAs to the 5′UTR regions, inducing either activation or repression of translation. The miRNAs synthesis is presented in Figure 1.

Numerous studies focusing on tissue samples have demonstrated the role of miRNA in OC, characterized by a wide-scale deregulation of miRNAs and aberrant expression of miRNAs correlated with histotype, histological grade, lymphovascular space involvement, lymph node and distant metastasis as well as FIGO stages [34,35,36,37]. Dhar Dwivedi et al. reported 53 miRNAs upregulated and 68 miRNAs downregulated in OC. In the upregulated miRNA group, a total of 7605 gene targets were found. Among them, miRNA-20a-5p and miRNA 106a-5p regulate 14.1% and 9.4% of target genes, respectively. Kyoto Encyclopedia of Genes and Genomes (KEGG) pathway analysis enrichment was performed for these upregulated miRNA target genes, identifying 67 and 24 pathways as enriched. Similarly, for downregulated miRNAs, a total of 9287 gene targets were identified. miR-26b-5p, miR-519d, miR-15a, and miR-15b regulated respectively 20.8%, 11.3%, 8.6%, and 8.9% of the target genes, with 41, 95, and 38 enriched pathways [38].

In contrast to the miRNA expression extensively analyzed in tissue samples, relatively little data are available on their expression in biofluids. This point is particularly important as it could allow preoperative tumor assessment, improving therapeutic strategy and decision making. There is published evidence that serum miR-221 [39], mir-205 [40], mir-375 [41], mir-210 [42], mir-34a-5p [43], mir-92 [44], mir-93 [43], mir-141 [45] mir-7 [46] and mir-429 [47] are upregulated in the biofluids of patients with different types of ovarian cancers. On the other hand, expression of microRNAs let-7f [48], mir-93 [45], mir-199a [49] and mir-148a [50] is downregulated in the biofluids of patients with ovarian tumors.

From the diagnostic point of view, Oliveira et al. [51] evaluated the profile of plasma miRNAs on a panel of 46 candidates, finding four upregulated miRNAs (miR-200c-3p, miR-221-3p, miR-21-5p and miR-484) and two downregulated (miRNA-195-5p and miRNA-451a). However, only two miRNAs (miRNA-200c-3p and miRNA-221-3p) were confirmed in a validation cohort. Savolainen et al. [52], on a short series of nine patients, found that miRNA-200a, miRNA-200b and miRNA-200c, in both tumor tissue and plasma, allowed discrimination between malignant and benign samples. In addition, a correlation was found between the expression of miRNA-200 in urine and plasma with the malignant status of tumors. Another study reported a higher level of miR-590-3p in OC plasma compared to a control group [53]. Chang et al. [54] observed for germ cell tumors of the ovary (OGCT) and sex cord (SCST)-specific expression profiles of miRNAs in nine OGCTs (two malignant and seven benign) and three SCST. Overexpression of miRNA-373-3p, miRNA-372-3p and miRNA-302c-3p and underexpression of miRNA-199a-5p, miRNA-214-5p and miRNA-202-3p were reproducibly observed in malignant OGCT versus benign OGCT or SCST. Yokoi et al. reported a plasma signature composed of six miRNAs selected after RT-qPCR (miRNA-200a-3p, miRNA-766-3p, miRNA-26a-5p, miRNA-142-3p, let-7d-5p and miRNA-328 -3p) able to successfully distinguish patients with ovarian cancer from healthy controls (AUC: 0.97; sensitivity, 0.92; and specificity, 0.91), paving the way for screening ovarian cancer [55]. Among five miRNAs, Zhu et al. observed that only serum miRNA-125b could distinguish benign controls and EOC patients [56]. Moreover, among the miR-200 family, Meng et al. [57] identified that serum levels of miRNA-200a (*p* = 0.0001), miRNA-200b (*p* = 0.0001), and miRNA-200c (*p* = 0.019) could distinguish benign from malignant ovarian tumors. Resnick et al. [58], confirmed that miRNAs in serum could be used as a marker for ovarian cancer in a series of 28 patients, based on over-expression of miRNA-21, miRNA-92, miRNA -29a, miRNA-93 and miRNA-126, and underexpression of miRNA-99b, miRNA-127 and miRNA-155. In a meta-analysis on the diagnostic value of serum miRNA-21 expression, including six studies with limited sample size, Qiu & Weng reported a pooled respective sensitivity, specificity, and AUC of 0.81 (95%CI: 0.73–0.88), 0.82 (95%CI: 0.75–0.87), and 0.89 (95%CI: 0.85–0.91) imposing further validation [59]. Recently, Wenyu Wang developed a plasma signature for malignant tumors using extracellular vesicles, and identified a panel of eight miRNAs (miR-1246, miR-1290, miR-483, miR-429, miR-34b-3p, miR-34c-5p, miR-145–5p, miR-449a). Their model had a respective AUC of 0.9762 and 0.9375 in the training and the validation set [60]. In contrast to several studies focusing on serum miRNAs, Kai Berner et al. focused on urinary expression of twelve microRNAs. In their experience, miR-15a was upregulated whereas let-7a was downregulated in ovarian cancer patients [61].

From the prognostic point of view, Zhu et al. [56] found that elevated serum miRNA-125b levels were higher in patients with early OC stages (FIGO stages I-II), and with no residual tumor after surgery. In addition, elevated serum miR-125b were correlated with progression-free survival (*p* = 0.035). Meng et al. reported that high levels of miRNA-200b and miR-200c were associated with poorer overall survival (*p* = 0.007, *p* = 0.017, respectively) [57]. Gao et al. [62], in a study including 74 serum samples of OC patients, 19 of borderline tumors and 50 of healthy controls, found that elevated serum miRNA-200c was correlated with improved two-year survival, while decreased serum miR-145 levels was associated with disease progression [63]. Finally, Zuberi et al. observed in OC an association between high expression of serum miR-125b and lymph node and distant metastases [64].

From the theragnostic point of view, most studies are based on cell cultures [65,66]. Yang et al. [67] found that protein kinase B (AKT) pathway activation was regulated by miRNA-214 and miRNA-150. Moreover, Echevarria-Vargas et al. reported a cisplatin resistance associated with miR-21 expression [68]. Lu et al. [69] observed that let-7a expression was significantly lower in OC patients sensitive to platinum and paclitaxel compared to those resistant to these agents. Langhe et al. reported that a panel of four miRNAs (let-7i-5p, miRNA-122, miRNA-152-5p and miRNA-25-3p) significantly downregulated in OC with potential contribution to drug resistance [70]. Finally, in a recent review, Saburi et al. [71] noted a relation between miRNA-30a-5p, miRNA-34a, miRNA-34a-5p, miRNA-98-5p, miRNA-142-5p, miRNA-338-3p, miRNA-708 and cisplatin resistance. Similarly, a relation was noted between miRNA -136, miRNA-338-5p, miRNA-503-5p, miRNA-1246, miRNA-1307 and paclitaxel resistance. Finally, a relation was noted between miRNA509-3p and platinum resistance and between miRNA-744-5p and carboplatin resistance. However, it is important to note that none of these miRNAs have been evaluated in biofluids, which could be a major contributor to adapting chemotherapy.

From the analysis of the literature on miRNAs in biofluids, it appears that there are arguments to suggest their role in physiopathology, in the differential diagnosis between benign and malignant tumors and to a lesser degree with borderline tumors, as well as to support their diagnostic, prognostic and theragnostic values. However, these analyses were mainly performed by microarray with validation by RT-qPCR, with potential biases linked to the methodology as proven in the context of endometriosis [72]. Moreover, the small number of studies with limited sample size focusing on the specific evaluation of miRNAs expression in biofluids of patients with ovarian tumors limit their potential clinical utility. Indeed, Langhe et al. [70] point out that miRNAs are abundant in tissues but are often rare in plasma and serum. For the quantification of miRNAs in plasma, the authors stressed that it was essential to use a high-sensitivity platform such as Next Generation Sequencing (NGS). However, no studies using NGS and bioinformatic tools to analyze miRNA content in large blood, serum, urine or saliva series are available to determine their role in routine practice.

## 3. PiRNAs

piRNAs are small ncRNAs of 24–32 nucleotides [73]. piRNA biosynthesis is summarized in Figure 2 [73,74]. Dysregulation of piRNAs and proteins (e.g., PIWI family proteins) has been observed in various cancers including OC [74,75]. The main function of piRNAs is to protect the genome from transposons. Giulio Ferrero et al. analyzing piRNA expression in urine, plasma exosomes, and stool observed that urine samples exhibited the highest piRNAs expression [76]. The piRNAs production and function are presented in Figure 2.

Singh et al. [77] found that piRNAs distinguish endometrioid from serous OC with 159 and 143 piRNAs differentially expressed, respectively. Among these piRNAs, 74 were upregulated and 77 downregulated in endometrioid OC, and 56 upregulated and 81 downregulated in serous OC. piR-52,207 was found to be upregulated in endometrioid OC, and both piR-52,207 and piR-33,733 in serous. Interestingly, among 20 biofluids evaluated by Hulstaert et al. [78], saliva has the highest fraction of piRNAs. Thus far, in the specific setting of OC, little data are available on piRNA expression in blood samples allowing for the evaluation of their potential diagnostic and prognostic values, and no study has evaluated saliva piRNA expression.

## 4. Transfer-RNAs (tRNAs)

Transfer RNAs (tRNAs) are a source of small regulatory RNAs (tsRNAs) acting on protein translation [38,79]. Based on the cleavage site, tsRNAs are divided into transfer RNA-derived RNA fragments (tRFs) and tiRNAs [80] with tumorigenesis functions [81,82,83,84,85,86,87]. tRFs are also involved in gene expression, oncogene activation and ovarian cancer progression through association with Ago and PIWI proteins [88].

Dhar Dwivedi et al. [38] observed that tsRNAs can predict abnormal cell proliferation with high accuracy in serum samples from a cohort of patients, healthy controls and benign and malignant tumors [89]. Eric Y. Peng observed that four tsncRNAs differentially expressed in serum samples had a high diagnostic accuracy for malignancy with an AUC of 0.95. Similarly, serum tRF-03357 and tRF-03358 levels are increased in patients with high-grade OC [90,91]. In addition, i-tRF-GlyGCC is linked to advanced FIGO stages, suboptimal debulking and, most importantly, with early progression and poor overall survival in EOC patients [91].

Despite a relatively abundant literature on tRNA, no study has focused on their diagnostic, prognostic and theragnostic relevance in the specific setting of OC.

## 5. Circular RNAs (CircRNAs)

CircRNAs are a large class of ncRNA with more than 70,000 specimen identified in human tissues [92,93,94,95,96,97,98] and in many cancerous cell types including OC [94,99,100,101]. CircRNAs Biosynthesis see Figure 3. circRNAs have a high prevalence, specificity [102,103], stability [104] and conservation [105], conferring a particular value as biomarkers. Moreover, using 20 biofluids, Hulstaert et al. demonstrated that circRNAs are enriched in biofluids compared to tissues [78]. Indeed, the median circRNA read fraction in biofluids was 84.4% vs. 17.5% in tissues. circRNAs act as specific miRNA reservoirs or sponges, as protein or peptide translators, and as regulators of gene transcription and expression, and interact with RNA-binding proteins (RBPs) impacting on the transcription and translation of genes.

From the diagnostic point of view, in the serum of OC patients, Wang et al. identified five circRNAs (circ-0002711; Chr5:170610175-170632616+; circ-0001756; Chr4:147227078-147230127-; and Chr16:53,175091-53191453+) with diagnostic value [106].

From the prognostic point of view, a previous study demonstrated that serum circ-0049116 released from a cell surface protein (mucin 16) could have value [107]. Increased expression of the circ-MUC16/miR-199a-5p axis positively correlates with the aggressiveness of OC. Expressions of circ-FAM35b, circ-051239, circ-ABCB10, circ-0072995, circ-EEF2, circ-RAB11FIP1, circ-FGFR3, circ-NOLC1 and circ-PGAM1 were correlated with metastasis of EOC [108,109,110,111,112,113,114,115]. Moreover, circ-0015756, circ-0002711, hsa-circ-0015326, circ-0001068, circ-0025033 and circ-KIF4A exhibited prognostic value in OC [116,117,118,119]. Serum circ-SETDB1 is positively correlated with lymph node metastasis and advanced stages of serous OC [120].

From the theragnostic point of view, circ-0002711/miR-1244/ROCK1 and has-circ-0015326/miR-127-3p/MYB pathways could be potential therapeutic targets [117]. CircRNAs, has-circ-0000714, circ-TNPO3 and circ-NRIP1 are expressed in OC with Paclitaxel resistance [121,122,123]. You et al. also reported that Circ-0063804 promoted OC cell proliferation and resistance to cisplatin by enhancing CLU expression via sponging miR-1276 [124]. In a recent review, Min Liu et al. [125] reported the different implications of circ-RNAs in the pathogenesis of OC, and underlined that circRNA-Cdr1as inhibited OC cell proliferation and promoted cisplatin-induced cell apoptosis, while circRNA-TNPO3 enhanced paclitaxel resistance. Despite the potential contribution of these circ-RNAs to the diagnostic, prognostic and theragnostic value in patients with OC, no study has focused on these biomarkers in biofluids in this specific setting.

## 6. Small Nucleolar RNAs (snoRNAs)

Small nucleolar RNAs (snoRNAs) are a class of non-coding RNAs with 60–300 nucleotides, and are mainly divided into two classes: C/D box SnoRNAs and H/ACA box SnoRNAs [126]. Most snoRNAs act as guide RNAs for the post-transcriptional modification of ribosomal RNAs, by modifying 2′-O-ribose methylation and pseudo-uridylation of ribosomal RNAs (rRNAs) [125]. Cumulative evidence demonstrates that snoRNAs play a role in the tumorigenesis of various cancers [127,128,129,130]. Using microarray on 197 EOC (162 serous, 15 endometrioid, 11 mucinous, and 9 clear cell), Oliveira et al. found that SNORA68 and SNORD74 were associated with decreased overall survival (OS) and poor clinicopathological features [131]. In an in vitro study Huilong Lin et al. observed that SNHG5 enhanced the sensitivity of ovarian cancer cells to paclitaxel by sponging miR-23a [132]. Wenjing Zhu et al. developed a signature based on nine snoRNAs (SNORD126, SNORA70J, SNORD3C, SNORA75B, SNORA58, SNORA11B, SNORA36C, SNORD105B, SNORD89,) to predict the prognosis of OC patients [133]. Finally, Peng-Fei Zhang et al. reported that SNHG22 overexpression is associated with poor prognosis and induces chemotherapy resistance to cisplatin and paclitaxel via the miR-2467/Gal-1 signaling pathway in EOC [134]. Although the number of dysregulated snoRNAs in ovarian cancer is up to 462 [127], one preliminary study investigated the role of snoRNA RNU2-1f in ovarian cancer. In this study, snoRNA abundance was investigated in serum (n = 10) by microarray analysis and validated in a serum set (n = 119) by reverse-transcription quantitative PCR. They reported that abundance of U2-1 snoRNA fragment (RNU2-1f) was significantly increased in sera of ovarian cancer patients (*p* < 0.0001) and paralleled International Federation of Gynecology and Obstetrics stage as well as residual tumor burden after surgery (*p* < 0.0001 and *p* = 0.011, respectively).

## 7. Long Non-Coding RNAs (lncRNAs)

lncRNAs act via various pathways to regulate gene expression at different levels [135] with a biogenesis similar to mRNAs (Figure 3). Arbitrarily, lncRNAs are defined as being composed of more than 200 nucleotides mainly between 1000 and 10,000.

Four different archetypes of lncRNA functions have been described (Figure 4) [136]. Since the discovery of lncRNAs, more than a thousand publications have been listed in PubMed in the specific setting of OC, although only about one percent focused on their expression in biofluids. Numerous studies about lncRNAs, mainly based on OC tissue samples, have demonstrated an association between clinicopathological characteristics such as histological type and grade, FIGO stages, lymph node and distant metastasis and some lncRNAs [137]. In a review including 34 studies involving more than 4000 women with OC, Hosseini and al observed an association between lncRNAs expression and PFS (HR: 1.88, 95% CI: (1.35–2.62)) and DFS (HR: 6.07, 95% CI: 1.28–28.78)). However, among the 34 studies, only one evaluating lncRNA in plasma [138] was included. They concluded that their work supported the robust prognostic significance of altered lncRNAs in ovarian cancer, but that more extensive studies are required.

From the diagnostic point of view, numerous studies have reported a relation between some lncRNAs and clinicopathological characteristics using OC samples analyzed by RT-qPCR, microarray and hybridization, and fluorescence in situ hybridization (FISH) from cancer tissue compared with adjacent normal tissue or samples from healthy patients (Salamini-Montemurri). In a recent review, Salamini-Montemurri et al. listed the various lncRNAs with clinicopathological value [137]. Among them, only E2F4AS [139], FLVCR1-AS1 [140], LINK-A [141], MLK7-AS1 [142] and aHIF [143] were evaluated in blood serum, but none of them exhibited a sufficient diagnostic value. Chun-Na Liu et al., in 185 EOC patients and 43 healthy volunteers, evaluated by RT-qPCR the expression of LOXL1-AS1 showing a higher expression in EOC patients with an AUC of 0.843 but a sensitivity and specificity of only 65.3% and 68.2%, respectively [144]. Using RT-qPCR, Jiezhi Ma & Min Xue investigated the expression of LINK-A in the plasma of 68 patients with OC and 34 healthy females, showing a higher level in OC patients. Recently, for the diagnosis of OC, Barwal and al found that blood lncRNA RP5-837J1.2 had a sensitivity, specificity and AUC of 97.3%, 94.6% and 0.99, respectively, but without external validation [145].

From the prognostic point of view, some studies have evaluated the value of lncRNAs in blood or serum to predict survival. In the meta-analysis using RT-qPCR, Chen et al. evaluated plasma levels of MALAT1 in 47 patients with EOC with metastasis (EOC/DM), 47 patients without metastasis (EOC/NDM), and 47 healthy controls (HC) [146]. Plasma MALAT1 allowed to distinguish EOC/DM and HC with an AUC of 0.884 (95% CI, 0.820–0.949; *p* < 0.001) with respective sensitivity and specificity of 89.4% and 72.3%. Jianming Gong et al., analyzing plasma samples from 66 patients with OC and 54 healthy controls, showed that lncRNA MIR4435-2HG was higher in patients with stage I-II FIGO stages OC, but with a high overlap of the value between the groups [147].

From the theragnostic point of view, Weiwei Xie et al. reviewed the contribution of lncRNAs in response to chemotherapy [148]. The main lncRNAs involved in the cisplatin resistance were HOTAIR, H19, MALAT1, MEG3, XIST, DNM3OS and ANRIL. Other lncRNAs have been proven to be associated with drug resistance, such as LSINCT5, NEAT1 for paclitaxel resistance, UCA1 for paclitaxel–cisplatin resistance, and GAS5 for platinum resistance. Until now, most of these lncRNA have merely been evaluated on very small series of plasma [149,150], not allowing to draw conclusions on their relevance.

## 8. Perspectives and Conclusions

Despite abundant literature on ncRNAs in OC, mainly based on cell culture and tissue samples, relatively few data are yet available on biofluid. Meanwhile, previous studies [78,151,152] have demonstrated the possibility of quantifying ncRNAs in various biofluids such as plasma, serum, urine and saliva. Moreover, it is important to note some limitations of the previously published studies, such as the small sample size, the absence of external validation, and the use in most studies of RT-qPCR and microarrays allowing ncRNA quantification of a predefined set of target sequences, while NGS and bioinformatics, representing an unbiased biomarker discovery method, is rarely used. To improve the preoperative diagnosis of ovarian cancer, studies evaluating the expression of ncRNAs in easily accessible biofluids should be promoted, imposing the use of new sequencing technologies. Thus far, to our knowledge, only two studies, the clinical trial NCT03738319 focusing on ncRNA profile in exosomes of OC patients, ref. [153] and the clinical trial NCT [154] evaluating the saliva expression of ncRNA in ovarian tumors including benign, borderline and ovarian cancer, are ongoing.

## Figures and Tables

**Figure 1 cancers-15-01539-f001:**
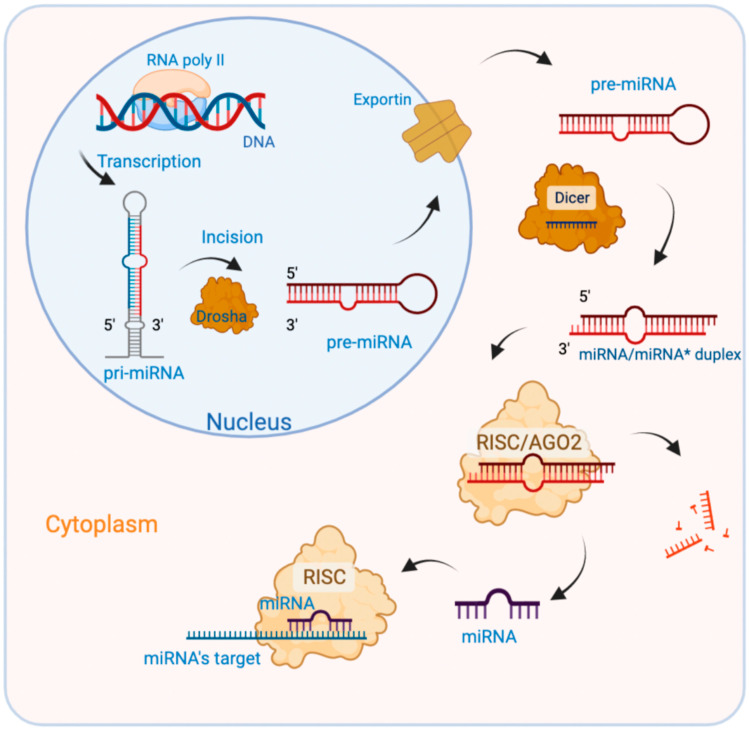
miRNAs synthesis. Biogenesis of miRNAs is a multistep process, beginning with the transcription of primary miRNAs (pri-miRNAs) by RNA polymerase II. The pri-miRNAs are transformed into precursor miRNAs (pre-miRNAs, 70 nucleotides long) by the RNase III Drosha-DGCR8-DDX5 microprocessor complex, and are then exported to the cytoplasm by Exportin (a Ran-GFP-dependent transporter). In the cytoplasm, pre-miRNAs are cleaved by the RNase Dicer-TAR RNA-binding protein (TRBP) complex, producing mature miRNA. Not all miRNAs pass through the canonical miRNA biogenesis pathway. Special miRNAs called mirtrons are produced from spliced introns with structural features such as pre-miRNAs and undergo a miRNA processing pathway that bypasses the Drosha-mediated cleavage step [32].

**Figure 2 cancers-15-01539-f002:**
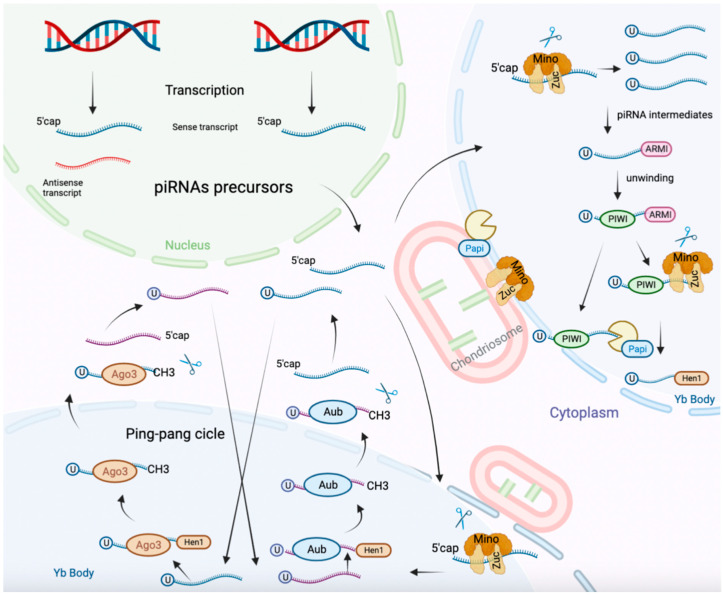
piRNAs synthesis and function. Mature piRNAs are generated from the processing of single-stranded RNAs transcribed from piRNA clusters in the genome, representing the largest class of non-coding RNAs found in most species. piRNAs can be also generated in the cytoplasm by a mechanism called the “ping-pong” cycle involving piRNA-directed antisense primary cleavage of transposon transcripts by Aubergine and PIWI proteins. piRNAs can modulate histone modifications and DNA methylation in a sequence-specific manner, leading to alterations in chromosomal conformation and gene expression regulation.

**Figure 3 cancers-15-01539-f003:**
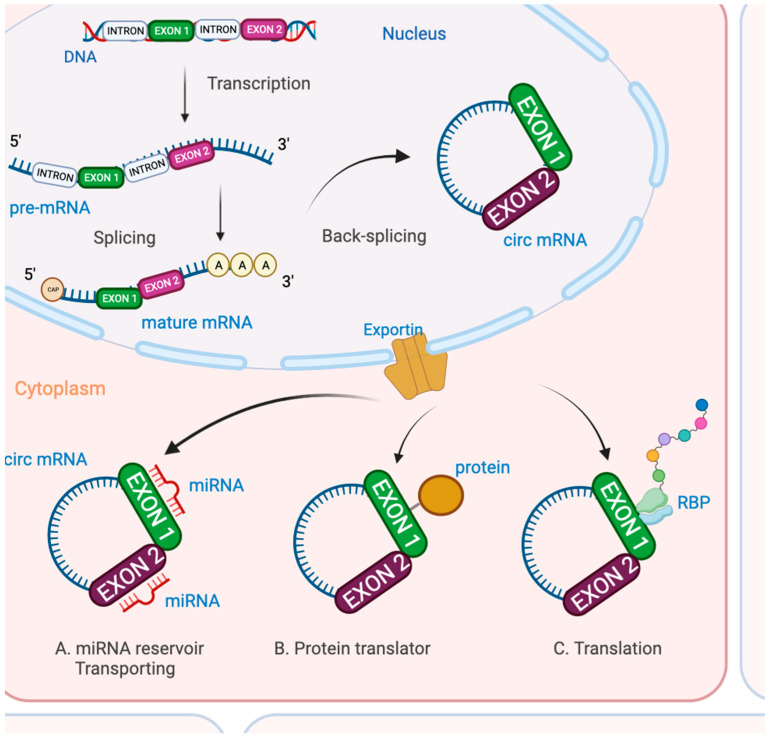
CircRNAs Biosynthesis. Biosynthesis of circular RNA (circ-RNA) are expressed from mostly protein-coding genes. This is a multistep process: firstly, the transcription of pre-messenger-RNA (mRNA), secondly, splicing into mature mRNA and then back-splicing into circ-RNA. Circ-RNA are then exported to the cytoplasm to (A) act as micro RNA (miRNA) reservoirs or transport miRNA; (B) act as protein or peptide translators; and (C) interact with RNA-binding proteins (RBPs) for transcription or translation of genes.

**Figure 4 cancers-15-01539-f004:**
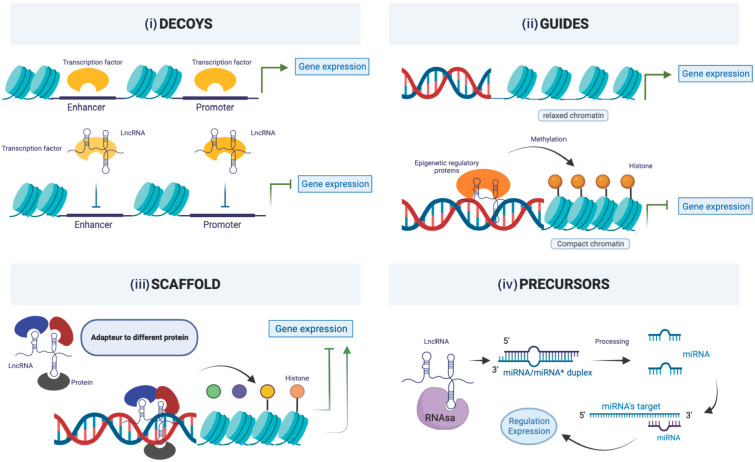
lncRNAs synthesis and functions. Four different archetypes of lncRNAs functions have been described; (i) lncRNAs can act as molecular signals occurring at a specific time and location to integrate developmental signals, interpret the cellular context, or respond to various stimuli, (ii) lncRNAs act as a molecular decoy by binding to their target proteins following transcription with mainly negative regulation, (iii) lncRNAs act as gene expression guides under a cis-form in the immediate vicinity of genes, and under a trans-form for distant genes, and (iv) lncRNAs act as scaffolds.

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
