# Peer review of "Insight on Non-Coding RNAs from Biofluids in Ovarian Tumors"

_cancers, 2023, doi:10.3390/cancers15051539_

Round 1

Reviewer 1 Report

In the present work Dabi et al. present a comprehensive review of the potential use of non-coding (nc) RNAs as biomarkers in the biofluids of patients with ovarian cancer. The authors review research articles focusing on how the expression of different types of ncRNAs (including microRNAs, pRNAs, tRNAs, circular RNAs, snoRNAs and lncRNAs) is altered in biolfuids of patients suffering from ovarian tumours. Overall, the manuscript is well written, it unifies data from studies using high-impact journals as references and is impactful to the field of oncology. I have the following comments regarding the review presented.

Minor points

1.     The correction of the following typos and language (indicated in bold) is strongly advised. 

Line 19: is the most lethal type of cancer

Line 34-35: Non-coding RNAs are classified

Line 39: the goal of the present work is to offer

Line 59: During their lifetime

Line 67: Except from

Line 82: CA125 is the most used biomarker

Line 109: miRNAs

Line 118: tissue not tissues

Line 126: target not targets

Line 133: decisions not decision

Line 177: were not was

Line 198: analyses not analyzes

Line 206: bioinformatic tools

Line 209: piRNAs

Line 210: piRNA biosynthesis

Line 230: Trasfer RNAs (tRNAs)

Line 245: CircRNAs

Line 246: CircRNAs are 

Line 272: reported the different implications instead various function

Line 278: snoRNAs

Line 287: In an in vitro

Line 296: LncRNAs

Line 297: act via various

Line 300: archetypes of lncRNA

Line 303: Numerous studies about lncRNAs

Line 344: and instead of et

Line 347: demonstrated instead of demonstrating

Line 351: bioinformatics instead of bioinformatic 

2.     Add descriptions of Figures 1, 2 and 3

3.     Lines 308-309: it is mentioned that among 34 studies only one evaluating lncRNAs in plasma was including. It would be interesting to briefly summarize and include in the review the conclusions of this study.  

4.     Line 316: Salamini-Montemurri. This reference (included in brackets) does not follow the same citation style as the rest

Major points

1.     There is a very large literature on the role of microRNAs in biofluids of ovarian cancer patients that could be considered and cited in order to place the current review in context. 

There are published evidence that serum miR-221(1), mir-205(2), mir-375(3), mir-210(4), mir-34a-5p(5), mir-92(6), mir-93(7), mir-141(8) mir-7(9) and mir-429(10) are upregulated in biofluids of patients with different types of ovarian cancers. The role of mir1-41 is discussed in reference 50, while the role of mir-373 in reference 45 of the manuscript. On the other hand expression of microRNAs let-7f(1), mir-93(2), mir-199a(3) and mir-148a(4) is downregulated in biofluids of patients with ovarian tumours. 

Finally, the implication of snoRNA RNU2-1f, which is upregulated in ovarian cancer patients, should be included in the manuscript. 

Author Response

First of all, we would like to thank again the reviewers on their time spent reviewing our work, and for the very constructive and relevant comments. All comments were considered carefully. A detailed response has been formalized for each of them in this document. We hope that these substantial changes, considering their relevance and their clinical impact, will find a favorable issue.

Reviewer 1

In the present work Dabi et al. present a comprehensive review of the potential use of non-coding (nc) RNAs as biomarkers in the biofluids of patients with ovarian cancer. The authors review research articles focusing on how the expression of different types of ncRNAs (including microRNAs, pRNAs, tRNAs, circular RNAs, snoRNAs and lncRNAs) is altered in biolfuids of patients suffering from ovarian tumours. Overall, the manuscript is well written, it unifies data from studies using high-impact journals as references and is impactful to the field of oncology. I have the following comments regarding the review presented.

Minor points

  1. The correction of the following typos and language (indicated in bold) is strongly advised. 

Line 19: is the most lethal type of cancer

Line 34-35: Non-coding RNAs are classified

Line 39: the goal of the present work is to offer

Line 59: During their lifetime

Line 67: Except from

Line 82: CA125 is the most used biomarker

Line 109: miRNAs

Line 118: tissue not tissues

Line 126: target not targets

Line 133: decisions not decision

Line 177: were not was

Line 198: analyses not analyzes

Line 206: bioinformatic tools

Line 209: piRNAs

Line 210: piRNA biosynthesis

Line 230: Trasfer RNAs (tRNAs)

Line 245: CircRNAs

Line 246: CircRNAs are 

Line 272: reported the different implications instead various function

Line 278: snoRNAs

Line 287: In an in vitro

Line 296: LncRNAs

Line 297: act via various

Line 300: archetypes of lncRNA

Line 303: Numerous studies about lncRNAs

Line 344: and instead of et

Line 347: demonstrated instead of demonstrating

Line 351: bioinformatics instead of bioinformatic 

Thank you for this comment. All typos were corrected in the revised version.

  1. Add descriptions of Figures 1, 2 and 3

As suggested, we added the description of the figures that we removed in the first version to respond to the editor.

  1. Lines 308-309: it is mentioned that among 34 studies only one evaluating lncRNAs in plasma was including. It would be interesting to briefly summarize and include in the review the conclusions of this study.  

We improved the revised version and included the conclusion of the review aforementioned.

  1. Line 316:Salamini-Montemurri. This reference (included in brackets) does not follow the same citation style as the rest

This was corrected in the revised version of the manuscript.  

Major points

  1. There is a very large literature on the role of microRNAs in biofluids of ovarian cancer patients that could be considered and cited in order to place the current review in context. 

There are published evidence that serum miR-221(1), mir-205(2), mir-375(3), mir-210(4), mir-34a-5p(5), mir-92(6), mir-93(7), mir-141(8) mir-7(9) and mir-429(10) are upregulated in biofluids of patients with different types of ovarian cancers. The role of mir1-41 is discussed in reference 50, while the role of mir-373 in reference 45 of the manuscript. On the other hand expression of microRNAs let-7f(1), mir-93(2), mir-199a(3) and mir-148a(4) is downregulated in biofluids of patients with ovarian tumours. 

Finally, the implication of snoRNA RNU2-1f, which is upregulated in ovarian cancer patients, should be included in the manuscript. 

We thank the reviewer for this valuable comment and added these data in the revised version of the manuscript.

Reviewer 2 Report

Dabi et al. described current knowledges of various kinds of ncRNAs on clinical importance in ovarian tumors. Among them, authors reviewed the characteristics and clinical values of miRNA, piRNA, tRNA, circRNA, snoRNA, and lncRNA. Authors proposed that these ncRNAs in body fluids will be very useful tools for early and noninvasive diagnosis of ovarian cancer for obstetrician and gynecologist, if simple and rapid methods are established. I totally agree with author’s opinion and this review is suitable for this journal, however, there are few points that should be revised.                  

1.      Authors illustrates summaries of ncRNA’s biosynthesis and functions as Figures. For beginners of RNA biology (especially noncoding RNA), these figures are very clear to understand. But figure of circRNA has not been shown. It should be also illustrated and added to this manuscript.

2. Line 112; Is the word "5'UTR regions" true? I am worring that the correct one was "3'UTR".

Author Response

First of all, we would like to thank again the reviewers on their time spent reviewing our work, and for the very constructive and relevant comments. All comments were considered carefully. A detailed response has been formalized for each of them in this document. We hope that these substantial changes, considering their relevance and their clinical impact, will find a favorable issue.

Reviewer 2

Dabi et al. described current knowledges of various kinds of ncRNAs on clinical importance in ovarian tumors. Among them, authors reviewed the characteristics and clinical values of miRNA, piRNA, tRNA, circRNA, snoRNA, and lncRNA. Authors proposed that these ncRNAs in body fluids will be very useful tools for early and noninvasive diagnosis of ovarian cancer for obstetrician and gynecologist, if simple and rapid methods are established. I totally agree with author’s opinion and this review is suitable for this journal, however, there are few points that should be revised.                  

  1. Authors illustrates summaries of ncRNA’s biosynthesis and functions as Figures. For beginners of RNA biology (especially noncoding RNA), these figures are very clear to understand. But figure of circRNA has not been shown. It should be also illustrated and added to this manuscript.

As requested, a novel figure was added to summarize CircRNAs biosynthesis;

  1. Line 112; Is the word "5'UTR regions" true? I am worring that the correct one was "3'UTR".

Thank you for your careful reading. It is indeed 5’UTR that could be an alternative in this specific case.
